# Generating Intuitive Fairness Specifications for Natural Language Processing

**Florian E. Dorner[1,2]**   **Momchil Peychev[1]**   **Nikola Konstantinov[1]**   **Naman Goel[3]**
**Elliott Ash[1]**   **Martin Vechev[1]**
[1]ETH Zurich    [2]MPI for Intelligent Systems, Tübingen    [3]University of Oxford

## Abstract

Text classifiers have promising applications in high-stake tasks such as resume screening and content moderation. These classifiers must be fair and avoid discriminatory decisions by being invariant to perturbations of sensitive attributes such as gender or ethnicity. However, there is a gap between human intuition about these perturbations and the formal similarity specifications capturing them. While existing research has started to address this gap, current methods are based on hardcoded word replacements, resulting in specifications with limited expressivity or ones that fail to fully align with human intuition (e.g., in cases of asymmetric counterfactuals). This work proposes novel methods for bridging this gap by discovering expressive and intuitive individual fairness specifications. We show how to leverage unsupervised style transfer and GPT-3's zero-shot capabilities to automatically generate expressive candidate pairs of semantically similar sentences that differ along sensitive attributes. We then validate the generated pairs via an extensive crowdsourcing study, which confirms that a lot of these pairs align with human intuition about fairness in toxicity classification. We also show how limited amounts of human feedback can be leveraged to learn a similarity specification.

## 1   Introduction

Text classifiers are being employed in tasks related to automated hiring [1], content moderation [2] and reducing the toxicity of language models [3]. However, they were shown to exhibit biases based on sensitive attributes, e.g., gender [4] or demographics [5], even for tasks in which these dimensions should be irrelevant. This can lead to unfair decisions, distort analyses based on these classifiers, or propagate undesirable stereotypes to downstream applications. The intuition that certain demographic indicators should not influence decisions can be formalized in terms of *individual fairness* [6], which posits that *similar inputs* should be *treated similarly*. In a classification setting we assume similar treatment for two inputs to require both inputs to be classified the same, while the notion of input similarity captures the intuition that certain input characteristics should not influence model decisions.

**Key challenge: generating valid, intuitive and diverse fairness constraints**   A key challenge for ensuring individual fairness is defining the similarity notion $\phi$, which can often be contentious, since fairness is a subjective concept, as well as highly task dependent [6, 7]. In text classification, most existing works have cast similarity in terms of word replacement [5, 8–10]. Given a sentence $s$, a similar sentence $s'$ is generated by replacing each word in $s$, that belongs to a list of words $A_i$ indicative of a demographic group $i$, by a word from list $A_{i'}$, indicative of another group $i' \neq i$. This approach has several limitations: (i) it relies on exhaustively curated word lists $A_i$ of sensitive terms, (ii) the expressivity of the generated pairs is limited to word replacements, and (iii) many terms are only indicative of demographic groups in specific contexts, hence directly replacing them with other terms will not always result in a similar pair $(s, s')$ according to human intuition. Indeed,

2022 Trustworthy and Socially Responsible Machine Learning (TSRML 2022) co-located with NeurIPS 2022.

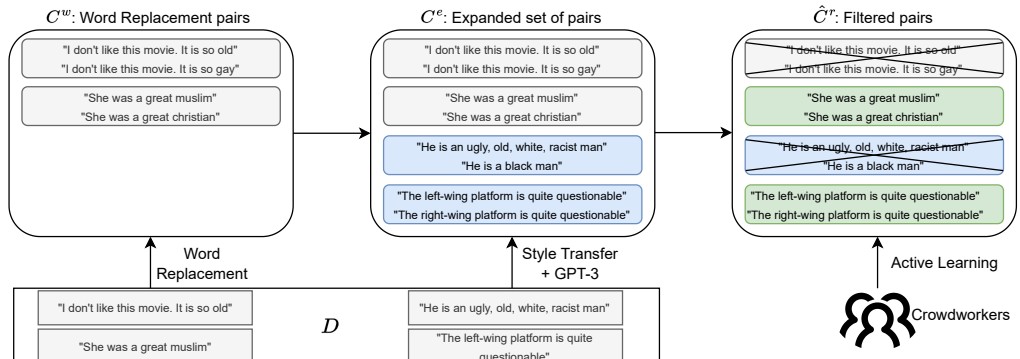

Figure 1: Workflow overview. We begin by generating sentence pairs using word replacement, and then add pairs of sentences leveraging style transfer and GPT-3. Then, we use active learning and crowdworker judgments to identify pairs that deserve similar treatment according to human intuition.

word replacement rules can often produce sentence pairs that differ in an axis not relevant to fairness (e.g., by replacing "white house" with "black house"). In addition, they can generate *asymmetric counterfactuals* [5]: sentence pairs $(s, s')$ that look similar but do not warrant similar treatment. For example, in the context of toxicity classification, the text "The movie is so old" may not be considered toxic while "The movie is so gay" clearly is.

**This work: generating fairness specifications for text classification**  The central challenge we consider in this work is generating a diverse set of input pairs that aligns with human intuition about which inputs should be treated similarly in the context of a fixed text classification task. We address this challenge via a three-stage pipeline (Fig. 1). First, we start from a dataset $D$ and generate a set $C^w$ of candidate pairs $(s, s')$ by applying word replacement to sentences $s \in D$. Second, to improve the diversity of pairs, we extend $C^w$ to a larger set $C^e$ by borrowing ideas from unsupervised style transfer. We change markers of demographic groups, e.g., "women" or "black people" in sentences $s \in D$ by replacing the style classifier in modern unsupervised style transfer methods [11, 12] with a classifier trained to identify mentions of demographic groups. In addition, we add pairs from GPT-3 [13], prompted to change markers of demographic groups for sentences in $D$ in a zero-shot fashion. Finally, to identify which of the generated pairs align with human intuition about fairness, we design a crowdsourcing study in which workers are presented with candidate pairs and indicate if the pairs should be treated similarly for the considered classification task or not. We employ active learning similar to [14] to train a BERT-based [15] classifier $\hat{\varphi}$ to recognize pairs that should be treated similarly using a limited amount of human feedback and obtain a filtered set of pairs $\hat{C}^r \subseteq C^e$. Our pipeline can be used in the context of most text classification tasks and in this work we instantiate it in the context of toxicity classification using a large dataset for online content moderation.

**Main contributions**  We make the following contributions: (i) we introduce a method for generating datasets of diverse candidate pairs for individual fairness specifications, leveraging GPT-3 and unsupervised style transfer to modify demographic attributes mentioned in sentences; (ii) we show that human feedback can be used to train a classifier which automatically identifies pairs that align with human fairness intuitions for a considered downstream task; (iii) we instantiate our framework in the context of toxicity classification, demonstrating that the proposed pairs are more diverse than word replacement pairs only and that crowdsourcing workers agree with more than $75\%$ of them.

## 2  Related Work

**Bias in NLP**  Early work on bias in NLP has focused on unwanted correlations between the word embeddings of identifiers for protected demographic groups and unrelated categories such as occupations [16, 17]. Recently, language models have been found to harbor stereotypical biases [10, 18–20]. Specific to text classification, identity terms such as "gay" and explicit indicators of gender have been shown to impact the outputs of classifiers trained to identify toxic comments [8] or to predict a person's occupation from their biography [4]. Olteanu et al. [21] demonstrate that human

perceptions of the quality of a toxicity classifier can depend on the precise nature of errors made by the classifier, as well as the annotators' previous experiences with hate speech. Blodgett et al. [22] recommend authors to explictly consider why, how and to whom the biases they identify are harmful.

**Language models for data augmentation**    Ross et al. [23] automatically create contrast sets [24] with a language model perturbing sentences based on control codes, while Rios [25] use style transfer to change the dialect of African-American Vernacular English tweets to Standard American English to evaluate the sensitivity to dialect of toxicity classifier. Hartvigsen et al. [26] use language models to generate a balanced dataset of benign and toxic comments about minority groups to combat classifiers' reliance on spurious correlations between identity terms and toxicity. Meanwhile, Qian et al. [27] train a perturber model to imitate human rewrites of comments that modify mentions of demographic groups, and demonstrate that their perturber can be used to reduce demographic biases in language models. However, this approach is limited by its reliance on expensive human rewrites and is only used for perturbations along fixed demographic axes such as gender.

**Learning fairness notions from data**    Ilvento [28] provides an algorithm to approximate individual fairness metrics for $N$ datapoints in $O(N \log N)$ queries, which can be practically infeasible. Meanwhile, Mukherjee et al. [29] suggest training a classifier to predict binary fairness judgments on pairs $(s, s')$ in order to learn a fairness metric $\phi$, but restrict themselves to Mahalanobis distances on top of a feature representation $\xi(s)$, limiting their expressive power. In contrast to our work, these works do not validate their learned fairness notions with human feedback. To that end, Cheng et al. [30] present an interface to holistically elicit stakeholders' fairness judgments, whereas Wang et al. [31] aim to learn a bilinear fairness metric for tabular data based on clustering human annotations.

# 3    Method

This section presents our end-to-end framework for generating and filtering valid candidate pairs for individual fairness specifications. In Sec. 3.1 we expand on existing word replacement definitions of individual fairness in text classification [5] by implementing three different ways to modify markers of demographic groups mentioned in a sentence $s$. Then, in Sec. 3.2 we leverage human feedback to learn an approximate similarity function $\hat{\varphi}$ to identify a set of relevant constraints $\hat{C}^r \subseteq C^e$.

## 3.1    Expanding fairness constraints

**Word Replacement**    First, we enrich the word replacement method by using the extensive lists of words associated with different protected demographic groups presented in Smith et al. [20]. The pool of terms is substantially larger than the 50 identity terms from Garg et al. [5]. We modify markers of group $j$ in a comment $s$ by replacing all words on the respective list of words associated with group $j$ with words from the list associated with the target group $j'$.

**Unsupervised Style Transfer**    Second, we use an unsupervised style transfer approach based on prototype editing (see [32] for an extensive review) to transform markers of a demographic group $j$ in a sentence $s$ to markers of another demographic group $j'$, creating a new sentence $s'$. Prototype editing identifies markers $a$ of a source style $A$ in a text $s$, and substitutes them by markers $a'$ of a target style $A'$. Our approach leverages that modern prototype editing algorithms utilize saliency methods in combination with a style classifier to identify markers of style, and instead uses a RoBERTa-based [33] classifier $c$ trained to identify sentences that mention specific demographic groups $j$. Combining ideas from [11] and [12], we transform a sentence $s$ to mention demographic attribute $j'$ instead of $j$ by iteratively masking tokens with large impact on the likelihood $p_c(j|s_m)$ (initially starting with $s_m = s$) until we reach a certain threshold, and filling the masked tokens using a BART-based [34] group-conditioned generator $g(s_m, j')$ trained to fill masks in sentences about group $j'$.

The unsupervised style transfer approach is likely to reproduce terms encountered during training, helping it to pick up on rare demographic terms that are particular to its training distribution which can be chosen to equal the training distribution for downstream tasks. In addition, unlike concurrent work by Qian et al. [27], unsupervised style transfer only requires labels $y_j(s)$ indicating the mention of demographic group $j$ in a sentence $s$ rather than expensive human-written examples of demographic group transfer. This allows us to modify mentions of demographic groups across axes like gender, religion and race, rather than restricting ourselves to changes within these axes.

**GPT-3**  Lastly, we leverage GPT-3 [13] to transform markers of protected demographic groups. We consider three methods: using GPT-3 standard mode and GPT-3 edit mode to rewrite sentences mentioning group $j$ to mention group $j'$ in a zero-shot fashion, as well as postprocessing sentences generated by word replacement to fix logical and grammatical inconsistencies with GPT-3 edit mode.

To ensure that mentions of demographic group $j$ were indeed replaced by $j'$ going from $s$ to $s'$, we use the same group-presence classifier $c$ as for the unsupervised style transfer approach to heuristically identify successful group transfer and discard pairs $(s, s')$ for which group transfer failed, for all three of our approaches. Implementation details are described in App. C and App. E contains examples.

## 3.2  Learning the similarity function

In order to evaluate to what extend the proposed similarity criteria align with human intuition, we conduct a crowdsourcing study, described in more detail in Sec. 4, to obtain labels $\varphi(s, s')$ which indicate whether a pair $(s, s')$ should be treated similarly for the sake of individual fairness $(\varphi(s, s') = 0)$ or not $(\varphi(s, s') = 1)$. We train a BERT-based [15] probabilistic model $p_{\hat{\varphi}}(s, s')$ that predicts values of the similarity function $\varphi(s, s')$ for pairs $(s, s') \in C^e$, and approximate the similarity function $\phi$ as $\hat{\varphi}(s, s') := 1 \Leftrightarrow p_{\hat{\varphi}}(s, s') > t$ for a given classification threshold $t$. To make optimal use of costly human queries, we employ active learning when training the classifier $\hat{\varphi}$, selecting pairs to label based on the variation ratios $1 - \max_y p(y|x)$ with $p$ estimated similar to Grießhaber et al. [14], based on Dropout-based Monte-Carlo [35, 36] applied to our model's classification head. Concretely, we iteratively select new unlabeled training data $D_i \subset C^e \setminus \bigcup_{j<i} D_j$ with $|D_i| = 1000$, based on the variation ratios, query labels for $D_i$, and retrain $\hat{\varphi}$ on $D_i$. As different annotators can disagree about whether two sentences $s$ and $s'$ should be treated similarly, we use a majority vote for evaluation. Inspired by Chen et al. [37]'s approach for dealing with noise in crowdsourcing, we use a single human query per pair $(s, s')$ during active learning, and relabel pairs that are likely to be mislabeled after active learning has concluded. See App. D for more details. When learning $\hat{\varphi}$ is completed, we can define the set of filtered constraints $\hat{C}^r = \{(s, s') \in C^e : \hat{\varphi}(s, s') = 0\}$.

# 4  Experiments

In this section, we experimentally evaluate our framework. Our key findings are: (i) the pairs generated by our method are more diverse compared to word replacement pairs only (Sec. 4.2), while mostly aligning with human intuition about fairness (Sec. 4.3) and (ii) the underlying similarity function $\varphi$ can be approximated by active learning from human judgements (Sec. 4.4).

## 4.1  Dataset and setup

We focus on toxicity classification on the Jigsaw Civil Comments dataset [38]. The dataset contains around 2 million online comments $s$ with labels $y(s)$ indicating toxicity. We focus on a subset $D' \subset D$ with labels $A_j(s)$ that indicate the presence of group $j$ in $s$ for training our group-presence classifier $c$, and only consider comments $s$ that consist of at most 64 tokens. We construct a set $C^e$ of 100,000 constraints applying our different generation approaches to $D'$[1]. More details on the generation and exact composition of $C^e$, as well as example pairs $(s, s')$, can be found in App. C. Throughout this section, whenever we report fairness for a classifier $f$, we refer to the proportion of pairs $(s, s')$ in a test pool of similar pairs for which $f(s) = f(s')$ rather than $f(s) \neq f(s')$.

## 4.2  Diversity of generated fairness constraints

To validate that our candidate constraint set $C^e$ is more diverse than word replacement on its own, we train 4 different toxicity classifiers, using Counterfactual Logit Pairing (CLP) [5] to empirically enforce different constraint sets $C^e, C_1, C_2, C_3$. Here $C^e$ corresponds to the full constraint set, as described in Sec. 3.1. while the other constraint sets have the same size as $C^e$, but contain pairs generated by one method only. In particular, the pairs in $C_1$ were generated by word replacement using the 50 identity terms from Garg et al. [5][2], the pairs in $C_2$ were generated by word replacement, using the larger list of terms of Smith et al. [20], and the pairs in $C_3$ were derived by style transfer.

---

[1] $C^e$ contains 42.5K word replacement and style transfer pairs each, and a total of 15K GPT-3 pairs.

[2] We did not discard any pairs from $C_1$ based on the group-presence classifier $c$.

Table 1: Balanced accuracy and fairness for a RoBERTa-based classifier $f$ trained with CLP using different constraint sets for training. Results are averaged over 5 runs and $\pm$ indicates the difference from the upper/lower bound of a naive $95\%$ confidence interval assuming normally distributed errors.

| Training/Evaluation | BA | $\mathrm{WR}_{50}$ ($C_1$) | WR ($C_2$) | ST ($C_3$) | Full $C^e$ |
|---|---|---|---|---|---|
| Baseline | $88.4 \pm 0.1$ | $78.4 \pm 1.4$ | $81.3 \pm 1.5$ | $76.7 \pm 1.8$ | $78.5 \pm 1.5$ |
| CLP(5) $\mathrm{WR}_{50}$($C_1$) | $87.0 \pm 0.3$ | $\mathit{98.3 \pm 0.1}$ | $89.1 \pm 1.9$ | $86.3 \pm 1.9$ | $87.3 \pm 1.8$ |
| CLP(5) WR ($C_2$) | $87.2 \pm 0.1$ | $93.1 \pm 1.2$ | $\mathit{98.2 \pm 0.4}$ | $90.5 \pm 1.7$ | $92.9 \pm 1.2$ |
| CLP(5) ST ($C_3$) | $85.9 \pm 0.1$ | $\mathbf{95.3 \pm 0.4}$ | $\mathbf{97.1 \pm 0.3}$ | $\mathit{95.4 \pm 0.4}$ | $95.5 \pm 0.3$ |
| CLP(5) Full $C^e$ | $85.0 \pm 3.4$ | $\mathbf{95.5 \pm 0.9}$ | $\mathbf{97.8 \pm 0.6}$ | $94.9 \pm 0.9$ | $\mathit{95.7 \pm 0.8}$ |

Table 2: Human evaluation: Answers to questions about comment pairs $(s, s')$. The first number represents the fraction of the answer across all queries, while the second number (in brackets) represents the fraction of comment pairs for which the answer was the majority vote across 9 queries.

| Metric/Method | Word replacement | Style Transfer | GPT-3 |
|---|---|---|---|
| Unfair: Average American | 84.9 (97.5) | 84.6 (95.8) | 83.4 (95.0) |
| Unfair: Own Opinion | 85.9 (97.5) | 85.2 (96.2) | 83.2 (93.7) |
| Group Transfer | 89.3 (95.0) | 79.2 (85.4) | 81.9 (89.5) |
| Content preservation | 88.1 (100) | 79.2 (91.2) | 78.4 (87.9) |
| Same Factuality | 73.0 (84.1) | 76.2 (87.5) | 78.5 (89.1) |
| Same Grammaticality | 91.2 (99.1) | 92.9 (97.9) | 92.9 (98.3) |

We then cross-evaluate the performance of the 4 classifiers trained with these constraint sets in terms of their test-time fairness according to each of the 4 fairness criteria, and their balanced accuracy.

The results in Table 1 show that each classifier achieves high fairness when evaluated on the set of pairs corresponding to the constraints used during its training (numbers in italics) while performing worse on other constraint pairs. While this indicates that adherence to fairness constraints does not always generalize well across our generation methods, we note that training on style transfer pairs ($C^e$ or $C_3$) generalizes substantially better to $C_2$ than training on different word replacement pairs ($C_1$; see the numbers in bold). More details can be found in App. C.

### 4.3 Relevance of generated fairness constraints

To validate that the generated fairness contraints are relevant and intuitive, we conducted a human evaluation with workers recruited via Amazon MTurk. The workers were presented with pairs $(s, s')$ consisting of a comment $s$ from the Civil Comments dataset, as well as a modified version $s'$ and asked about whether they believe that the two comments should be treated similarly and whether they believed that the average American shared their opinion. Treatment was framed in terms of toxicity classification for the sake of content moderation, ensuring that we verify the relevance of the learned notions relevant to this specific task. The workers were also asked whether the demographic group was transferred correctly from a given $j$ to a given $j'$, whether the content of $s$ has been preserved in $s'$ apart from the demographic group transfer, and whether there are differences in factuality and grammaticality between $s$ and $s'$. We collected human feedback for a set $S$ containing a total of 720 pairs $(s, s')$ with 240 each being produced by our style transfer approach, GPT-3 in a zero-shot fashion, and word replacement using the list from [5] as for $C_1$. These 240 pairs per method were split into 80 pairs for each of the axes male↔female, christian↔muslim and black↔white. Each pair $(s, s')$ was shown to nine different workers. Further details can be found in App. B.

Table 2 shows that all three methods mostly produce relevant fairness constraints, according to a majority of annotators. At the same time, they generally successfully modify the mentioned demographic group, and preserve content, factuality and grammaticality. While word replacement generally performs better in terms of group transfer and content preservation, it only has a small advantage in terms of relevance to fairness, perhaps due to its worse performance in terms of factuality:

Table 3: Performance of differently trained classifiers $\hat{\varphi}$ on the test set $T$. Active learning classifiers are retrained 10 times on the last batch $D_6$. Results are averaged and $\pm$ indicates the difference from the upper/lower bound of a naive 95% confidence interval assuming normally distributed errors.

| Method | ACC | TNR | TPR | BA |
|---|---|---|---|---|
| Constant Baseline | 78.8 | **100.0** | 0.0 | 50.0 |
| Active Learning t=0.5 | $79.8 \pm 0.3$ | $97.2 \pm 0.3$ | $15.1 \pm 1.2$ | 56.1 |
| Active Learning + Relabel t=0.5 | $\mathbf{81.1 \pm 0.3}$ | $95.5 \pm 0.7$ | $28.6 \pm 2.2$ | 62.0 |
| Active Learning t=0.01 | $78.7 \pm 1.1$ | $87.5 \pm 2.1$ | $45.7 \pm 1.8$ | 66.6 |
| Active Learning + Relabel t=0.01 | $78.3 \pm 0.7$ | $86.8 \pm 1.5$ | $\mathbf{46.6 \pm 2.5}$ | **66.7** |

we found examples in which word replacement changed "white house" to "black house"; or Obama is referred to as "white" rather than "black". These pairs were not seen as fairness constraints by most annotators and judged badly in terms of preserving factuality. See B.1 for more detailed results.

## 4.4 Learning the similarity function

We employed our active learning approach to efficiently train a classifier $\hat{\varphi}$ from relatively few human judgments, with the goal of using it to identify pairs that represent actual fairness constraints on the remaining pool of candidates. We conducted 6 steps of active learning with 1000 queries each and discarded failed queries, ending up with a total of 5490 labeled pairs $((s, s'), \varphi(s, s'))$. Details on our model architecture and other hyperparameters can be found in App. D. We evaluate our learnt classifier on a test set $T$ consisting of 500 randomly selected pairs from $C^e$ for which five annotators were asked about the average American's fairness judgment.

Because 78.8% of the pairs $(s, s')$ in $T$ represented fairness constraints ($\varphi(s, s') = 0$) according to the majority of annotators, we report Balanced Accuracy (BA), in addition to standard accuracy (ACC) and the true positive and negative rates (TPR and TNR). Table 3 displays these metrics for classifiers resulting from our active learning method for different classification thresholds $t$ and with and without relabeling. We observe that $\hat{\varphi}$ performs substantially better than random, achieving BA of 66.7% when used with an aggressive classifier threshold $t$. The table also validates our relabeling approach: after observing that our classifier was biased towards predicting $\varphi(s, s') = 0$, we collected two additional labels for 500 pairs $(s, s')$ for which both the human and the predicted label were equal to zero ($\hat{\varphi}(s, s') = \varphi(s, s') = 0$), selected based on the variation ratios. 47% of these pairs received a majority vote of $\varphi(s, s') = 1$, showing that our approach correctly identified pairs that were likely to be mislabeled. For the balanced classification thresholds $t = 0.5$, retraining our classifier on the updated majority votes also substantially increased TPR at little costs to TNR.

According to a qualitative evaluation, most of the sentence pairs $(s, s')$ predicted to not represent fairness constraints ($\hat{\varphi}(s, s') = 1$) had the words "boy" or "man" replaced by terms denoting identity membership. Such sentence pairs were often not seen as fairness constraints by our annotators, as the inclusion of the identity term can be interpreted as aggressive or mocking. $\hat{\varphi}$ also successfully identified sentence pairs $(s, s')$ for which $s'$ was unrelated to $s$, that were sometimes produced by GPT-3, as not representing fairness constraints. Additional results and details can be found in App. D.

## 5 Conclusion

We proposed a framework for producing expressive and intuitive specifications for individual fairness in text classification. We experimentally demonstrated that our constraints are indeed more expressive than previous constraints based on word replacement and validated that most of the generated fairness constraints were relevant in the context of toxicity classification according to human annotators. In addition, we used active learning to demonstrate that human fairness judgments can be predicted using limited amounts of training data. In future work we plan to utilize the generated filtered constraints to train a fair downstream toxicity classifier with better trade-off between accuracy and fairness. Further work could explore approaches similar to ours for other NLP tasks with discrete outcomes beyond classification, as well as for evaluating robustness to other forms of intuitively relevant perturbations.

## Acknowledgements

We would like to thank Dan Hendrycks, Mislav Balunović, Afra Amini and Dominik Stammbach for helpful comments and discussions during early stages of this work. We would also like to thank the anonymous reviewers for their comments and feedback.

Florian Dorner is grateful for financial support from the Max Planck ETH Center for Learning Systems (CLS) received during part of this work. Nikola Konstantinov's contributions to this publication were made possible by an ETH AI Center postdoctoral fellowship.

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

# A  Ethical Considerations

Our human evaluation experiments involving workers from Mechanical Turk were reviewed and approved by the ETH Zurich Ethics Commission as proposal EK 2022-N-117. Workers on Mechanical Turk were warned that they might be shown offensive comments as part of our study and were able to opt out of participating in our study at any time. We also made sure that the per-task compensation was sufficiently high to result in a hourly compensation exceeding the US federal minimum wage. More details on our human evaluation experiments can be found in App. B.

While we believe that our results show that learning more precise fairness notions by involving human feedback is a very promising area of research, we caution against directly using our learnt similarity classifier $\phi$ for evaluating fairness in high-stakes real-world applications of toxicity classification. First, our results show that there is substantial disagreement between different survey participants about which pairs $(s, s')$ require equal treatment by a fair classifier. While resolving these disagreements via a majority vote is a natural choice, other approaches may be desired in some contexts (for e.g., enforcing equal treatment whenever at least one participant believes it is required). Second, our survey participants may have geographic biases and are neither direct stakeholders, nor experts in discrimination law and hate speech. Given that our learning approach shows promising signs of being able to improve upon existing fairness definitions despite large amounts of disagreement, which is likely to be less common for actual stakeholders and experts, we recommend using it in conjunction with fairness judgments provided by application-specific experts and stakeholders.

# B  Further Details on Human evaluation

In order to participate, workers had to live in the US and be above 18 years old in addition to being experienced with MechanicalTurk (having completed more than $5000$ HITs[3] and having a good reputation ($97\%$ acceptance rate across all of the worker's HITs). Workers were warned about the potentially offensive content of some of the comments show in the study by the following statement: "Please note that this study contains offensive content. If you do not wish to see such content, please withdraw from the study by leaving this website." and were also told that they could withdraw from the study at any later point: "You may withdraw your participation at any time without specifying reasons and without any disadvantages (however, you will not get paid for the current HIT in case you withdraw before completing it)".

After encountering a high prevalence of bots, malicious workers or workers that fundamentally misunderstood our task instructions during pilot experiments, we had workers pass a qualification test by providing correct answers for nine out of ten queries $\varphi(s, s')$ for pairs that were hand-designed to have a relatively obvious correct answer. We validated these hand-designed pairs in a separate experiment, querying workers about $\varphi(s, s')$ for 11 pairs $(s, s')$, and asking them to verbally explain each of their decisions, paying a total of $1.83. We only included hand-designed pairs in the qualification test if at least eight out of ten workers produced the intended answer during this experiment, and no worker brought forward convincing arguments against this answer being correct.

Workers were paid $0.83 for a battery of 11 queries $\varphi(s, s')$ for the Active learning and the test set $T$ and first $1.20, later $1.30 for 11 queries about all questions presented in Table 2. In both cases, workers had to explain one of their answers verbally, and one of the queries was a attention check pair $(s, s')$ with a relatively obvious correct answer constructed in the same way as for the qualification tests.

Blocks of queries with wrong answers to the attention check question or (partially) incoherent verbal explanations were manually reviewed, and thrown out in case we were not able to find evidence that the worker had correctly understood the task in which case the worker was not paid. Workers were made aware of this by the following statement in the task description: "To ensure workers pay attention, we include pairs of comments for which there is a clear correct answer at random positions. Please make sure to carefully read every comment and answer diligently, as your HIT will be rejected if your answer for one of these pairs of comments is wrong, in which case you will not receive a payment for the HIT. We will also ask you to produce an explanation for one of your answers. Please write at least one sentence and make sure your explanation is coherent with the answer you gave, as your HIT might be rejected otherwise in which case you will not receive a payment for the HIT.".

---

[3]Bundled tasks on MechanicalTurk for which a remuneration is received on completion

According to https://turkerview.com, a tool used by many professional workers on MechanicalTurk, we paid workers an average hourly rate of $16.17, clearly exceeding the US minimum wage. While this is likely an overestimate, as not all workers use turkerview, the hourly rate is so high, that we still exceed the minimum wage for workers taking twice as long as the average worker using turkerview.

Workers were introduced to the task as follows: "We are investigating fairness in toxic comment classification and are looking for pairs of comments with similar content that should be treated the same by a content moderator (if one of them was removed, the other one should be removed too). You will be shown pairs of comments and be asked whether the comments are similar in various ways and whether you would consider it unfair for a content moderator to remove one of the comments in each pair but not the other.". The queries read as follows for the first task:

- Without further context, would the average american consider it unfair or acceptable for a content moderator to delete one of the shown comments but not the other?
  - It would be unfair for one of the two comments but not the other to be deleted (Both comments ought to be treated the same way, no matter what)
  - If comment a) was deleted, comment b) should be deleted as well, but not the other way around. (Deleting comment b) would be more justifiable)
  - If comment b) was deleted, comment a) should be deleted as well, but not the other way around. (Deleting comment a) would be more justifiable)
  - It would be acceptable to delete either of the comments without deleting the other. (Treating the comments differently either way would be fine)

and as follows for the second one (Placeholders for group terms are presented in *Italics*. The answer summarized in Table 2 is highlighted in **bold**):

- Without further context, would you consider it unfair or acceptable for a content moderator to delete one of the shown comments but not the other?
  - **It would be unfair for one of the two comments but not the other to be deleted (Both comments ought to be treated the same way, no matter what)**
  - If comment a) was deleted, comment b) should be deleted as well, but not the other way around. (Deleting comment b) would be more justifiable)
  - If comment b) was deleted, comment a) should be deleted as well, but not the other way around. (Deleting comment a) would be more justifiable)
  - It would be acceptable to delete either of the comments without deleting the other. (Treating the comments differently either way would be fine)
- What do you expect the average American to answer to the previous question?
  - **It would be unfair for one of the two comments but not the other to be deleted (Both comments ought to be treated the same way, no matter what)**
  - If comment a) was deleted, comment b) should be deleted as well, but not the other way around. (Deleting comment b) would be more justifiable)
  - If comment b) was deleted, comment a) should be deleted as well, but not the other way around. (Deleting comment a) would be more justifiable)
  - It would be acceptable to delete either of the comments without deleting the other. (Treating the comments differently either way would be fine)
- Do you consider one of the two comments to be more factually correct than the other?
  - Comment a
  - Comment b
  - **Neither comment / too little context to tell**
- Do you consider one of the two comments to be more grammatically correct than the other?
  - Comment a
  - Comment b
  - **Neither comment**
- Is comment a) about *group a* and comment b) about *group b*?
  - **Yes**

- – No, comment a) is not about *group a*
- – No, comment b) is not about *group b*
- – No, neither

- • Apart from differences related to *group a* and *group b*, are both comments similar in terms of content?
    - – **Yes, they are almost the same.**
    - – They are somewhat similar, but differ in some additional details.
    - – There is an important additional difference between the comments' content

B.1 shows the results of the human evaluation on our test set $S$ split along the axis of attribute transfer, rather than generation method as in 2. Along with the results in Table 2 they show that despite the general agreement about the relevance of the generated fairness constraints, there is substantial disagreement between annotators when it comes to deviations from the most common answer across all comments. In all cases, the fraction of comments with majority vote equal to that answer is substantially higher than the overall fraction of these votes across all comments and annotators. The same is true for our set $T$ of 500 randomly selected pairs from $C^e$ for which we only asked about the average American's fairness judgment: $70.9\%$ of the annotations were $\varphi(s, s') = 0$, while the same was true for $78.8\%$ of the per-comment pair majority votes.

Table B.1: Human evaluation: Answers to questions about comment pairs $(s, s')$ grouped along demographic group transfers along different axes. The first number represents the fraction of the answer across all queries, while the second number (in the brackets) represents the fraction of comment pairs for which the answer was the majority vote across 9 queries.

| Metric/Method | male↔female | black↔white | christian↔muslim |
| --- | --- | --- | --- |
| Unfair: Average American | 83.5 (96.6) | 82.2 (94.5) | 87.2 (97.0) |
| Unfair: Own Opinion | 83.5 (96.6) | 82.4 (92.9) | 88.4 (97.9) |
| Group Transfer | 82.6 (91.6) | 81.6 (86.6) | 86.2 (91.6) |
| Content preservation | 84.9 (95.4) | 79.5 (92.0) | 81.3 (91.6) |
| Same Factuality | 75.3 (82.9) | 73.6 (85.0) | 78.8 (92.9) |
| Same Grammaticality | 90.5 (97.5) | 92.2 (98.3) | 94.3 (99.5) |

## C  Further details on style transfer

**Unsupervised style transfer**  To transform markers of demographic groups in sentences, we first finetune a Multi-headed RoBERTa-based [33] classifier $c$ to predict labels $y_j$ indicating the presence of markers of a demographic group $j$ from a list of protected demographic groups $J$ in a sentence $s$. We use the population labels ("Black", "Male", "Heterosexual", "Muslim",etc.) that are provided for a subset of the Civil comments dataset. The group-presence classifier $c$ is based on the roberta-base model, followed by a linear layer with 768 neurons applied to the output embedding of the first token only, a Tanh layer, another linear layer mapping to a single dimension, and a Sigmoid layer. We train $c$ for 3 epochs with a batch size of 16 and use the Adam optimizer [39] with learning rate 0.00001 to optimize the binary Cross Entropy loss, reweighed by relative label frequency in the dataset. Table C.1 shows the balanced accuracy on the test set for all demographic groups in the dataset. For our downstream applications of $c$, we restrict ourselves to the demographic groups for which the classifier $c$'s balanced accuracy is above $90\%$. Furthermore, we also exclude the group labeled "mental illness" because the word replacement lists we used lack a clear analogon.

Then, we finetune a BART-based [34] generator $g$ on a mask-filling task on the same data: For every data point $s$, we sample a group from the set of demographic groups $j$ mentioned in $s$, i.e. $\{j : y_j(s) = 1\}$, skipping sentences $s$ for which no group $j$ meets this criterion. Inspired by [11] we mask all of $s$'s tokens that have an above-average attention value for the 11th layer of the classifier $c$, merge consecutive mask tokens into one, and prepend the name of the sampled group $j$ to the masked sentence before fedding it to the generator $g$. The generator $g$ is then finetuned to reconstruct $s$ using token-wise Cross Entropy.

Table C.1: Balanced accuracies of the group-presence classifier $c$ for different labels

| Category | BA | Category | BA | Category | BA |
|---|---|---|---|---|---|
| Male | 96.5 | Christian | 96.6 | Physical disability | 54.9 |
| Female | 97.8 | Jewish | 98.9 | Intellectual disability | 54.3 |
| Transgender | 99.3 | Muslim | 98.9 | Mental illness | 98.3 |
| Other gender | 50.0 | Hindu | 98.2 | Black | 99.2 |
| Heterosexual | 98.1 | Buddhist | 99.2 | White | 99.5 |
| Homosexual | 99.3 | Atheist | 99.6 | Asian | 98.3 |
| Bisexual | 65.4 | Other religion | 50.0 | Latino | 96.6 |
| Other sexuality | 50.0 | Other disability | 50.0 | Other race | 55.5 |

The BART-based generator $g$ is trained starting from the pretrained facebook/bart-large model for a single epoch with batch size 4, again using Adam and a learning rate of 0.00001. For filling in masked sentences, we pick the completion with the largest difference in the classifier $c$'s pre-sigmoid activation for the target and source demographic groups $j'$ and $j$ among candidate sentences produced by a beam search generation using the generator $g$ with width 5.

To transfer an example $s$ from mentioning group $j$ to mentioning group $j'$, we follow [12] and iteratively mask the token for which masking reduces $p_c(y_j|x)$ the most, until we reach a threshold of $p_c(y_j|x) < 0.25$. We use this approach rather than the attention-based masking from [11] because of the lack of theoretical motivation for using attention to identify important features [40], and because attention scores are the same for all of our model's group-presence prediction heads, rather than specific to a particular group $j$.[4] Then, we prepend a verbal representation of label $j'$ to $s$ to form a prompt $p$, and generate a sentence $s'$ as $g(p)$.

**Word replacement** Our word replacement approach is based on the list of words provided in [20]: Given a sentence $s$ mentioning demographic group $j$ and a target attribute $j'$, we replace all words in $s$ that are on the list associated with $j$ with random words from the list associated with $j'$, replacing nouns with nouns and descriptors with descriptors whenever possible, and nouns with descriptors otherwise. The full list of words we used for word replacement is displayed in Table E.1.

**GPT-3** We accessed GPT-3 using OpenAI's API[5]. For our first approach, we used the "text-davinci-001" version of GPT3 in a zero-shot manner with the prompt: "Please rewrite the following sentence to be about $j'$ rather than $j$:" followed by a new line and the targeted sentence $s$. The second approach was based on the beta-version of GPT-3's editing mode [6]. Here, $s'$ is produced using the model "text-davinci-edit-001" with the instruction "Rewrite the text to be about $j'$ rather than $j$". Lastly, we used to same model in conjunction with word replacement: First, we generated a candidate sentence $s''$ using the procedure described in the word replacement section. Then, in order to fix issues caused by the context-blindness of the word replacement approach, we postprocessed $s''$ using "text-davinci-edit-001" with the instruction "Fix grammatical errors and logical inconsistencies" to produce $s'$. We used temperature = 0.7 and top_p= 1 in all our approaches and used max_tokens= 64 for "text-davinci-001" to control the length of the modified sentence $s'$.

**Post-filtering** For all three approaches, we performed a post-filtering step to reduce the prevalence of unsuccesful attempts at demographic group transfer in our set of constraints $C^e$. Given a pair $(s, s')$ of an original sentence and a modified version, we only include it in our set of constraints $C^e$, if the classifier probability $p_c(y_{j'}|s')$ for label $j'$ is below 0.5 and the classifier probability $p_c(y_j|s')$ for label $j$ is above 0.5.

As mentioned in Sec. 4.1, we attempt to produce modified comments $s'_{j'}$ mentioning group $j'$ for each $s$ in $D'$ for all demographic groups $j$ with $y_j(s) = 1$ and all possible target groups $j'$ for word replacement and style transfer. For GPT-3, we attempted a total of 75 generations for each of our three

---

[4]We used attention during the training of $g$, for which dropping out some tokens unrelated to $j$ is less problematic, in order to save resources.

[5]https://openai.com/api/

[6]https://openai.com/blog/gpt-3-edit-insert/

Table C.2: Amount of generated pairs $(s, s')$ per generation method.

| Generation Method | Total (Train) | Total (Test) | In $C^e$ (Train) | In $C^e$ (Test) |
|---|---|---|---|---|
| Word Replacement | 980667 | 331490 | 42500 | 10625 |
| Style Transfer | 681111 | 229883 | 42500 | 10625 |
| GPT-3 Zero-Shot | 6322 | 2139 | 6200 | 1550 |
| GPT-3 Edit Mode | 3704 | 1199 | 3500 | 875 |
| GPT-3 Postprocessing | 5330 | 1831 | 5300 | 1325 |

generation modes per axis pair of demographic groups $(j, j')$ and direction of group transfer, with the source sentences $s$ randomly selected among the sentences with label $j$ in $D'$. For constructing the secondary test set $S$, we attempted more generations for the axes male↔female, christian↔muslim and black↔white, homosexual↔heterosexual. The latter axis was left out of $S$ because we found that the rate of successful generations was too limited. We generated a maximum of 2250 attempts up until a total of 250 successful generations (post-filtering step passed) for GPT-3's zero-shot mode, a maximum of 750 until to a total of 100 successful generations for GPT-3's edit mode, and up until a total of 100 successful generations for GPT-3 based postprocessing of word replacement. Table C.2 shows the overall amount of generated pairs per method.

As an additional experiment to validate the increased diversity of our constraint set $C^e$ we train a similarity classifier[7] $\hat{\varphi}$, on $C^e$ to distinguish pairs $(s, s')$ generated by word replacement from pairs generated by style transfer or GPT-3. Training on $100000$ examples without label noise, we are able to achieve $91.6\%$ test accuracy on a balanced test set, suggesting that there is a meaningful difference between pairs generated by word replacement and the rest of the constraint candidates $C^e$.

# D  Further details on learning similarity functions

First, Proposition D.1 below establishes that robustness with respect to a binary similarity function $\varphi$, i.e. $\varphi(s, s') = 0 \Rightarrow f(s) = f(s')$, can fully capture the definition of individual fairness as Lipschitz-Continuity proposed by Dwork et al. [6] for deterministic classifiers $f$.

**Proposition D.1.** *Given a metric $d : X \times X \to \mathbb{R}$, a binary metric $d_b : Y \times Y \to \{0, 1\}$ and a constant $L > 0$, there exists a similarity function $\varphi : X \times X \to \{0, 1\}$ such that a function $f : (X, d) \to (Y, d_b)$ is Lipschitz-Continuous with constant $L$ if and only if $\varphi(x, x') \geq d_b(f(x), f(x'))$ for all $x, x' \in X$.*

*Proof.* Define $\varphi(x, x') := \mathbb{1}\{Ld(x, x') \geq 1\}$. Then whenever $d_b(f(x), f(x')) = 1$, we have $d_b(f(x), f(x')) = 1 \leq \varphi(x, x')$ if and only if $d_b(f(x), f(x')) \leq Ld(x, x')$. But if $d_b(f(x), f(x')) = 0$, the Lipschitz inequality is allways true. Now, assume that $f$ is not Lipschitz: Then, there exist $x, x' \in X$ such that $1 = d_b(f(x), f(x')) > Ld(x, x')$, implying $0 = \varphi(x, x') < d_b(f(x), f(x')) = 1$ □

We use a BERT-based classifier that acts on a pair $(s, s')$ by first tokenizing both $s$ and $s'$ and padding the token representation to a length of 64, concatenating these tokens and feeding the concatenated token representation into a pretrained bert-uncased-base model. We then apply a linear layer with dropout ($p = 0.1$) followed by a Tanh layer and a second linear layer with dropout ($p = 0.1$) to obtain single dimensional logits, to which a sigmoid layer is applied before computing the binary Cross Entropy loss and optimizing it using the Adam optimizer with learning rate 0.00001 applied to all model parameters. We use BERT rather than more modern models such as RoBERTa [33] and Deberta [41], as we have found it to clearly outperform them for our task, plausibly because BERT uses a next-sentence-prediction task during pretraining, which is structurally similar to our task of comparing two sentences. Table D.1 demonstrates the advantage of using BERT, as well as concatenating token representations rather than learning based on the difference between separately produced BERT features for both $s$ and $s'$. Unless stated otherwise, our Active Learning approach trains for five epochs on each queried block $D_i$ before selecting new data $D_{i+1}$ to label.

---

[7]Using the same architecture as for our active learning experiments described in App. D

Table D.1: Different architectures trained for one epoch on 5000 samples from a set of pairs $(s, s')$ generated using word replacement to distinguish demograpghic group transfer within the same category of gender and sexuality, race and religion vs across categories ($\varphi_2$). "Featurediff" uses a linear model applied to the difference of model features produced for the respective first tokens in $s$ and $s'$. "Bilinear" uses a bilinear model on top of these feature differences instead. "Merge" appends $s'$ to $s$ before tokenization and learns a linear model on top of the model features for this combined input. "Concat" operates similarly, but first tokenizes $s$ and $s'$ and pads both to $64$ tokens before feeding the concatenated tokens into the model. No dropout was used in the post-BERT layers for these experiments. All results averaged over 10 runs and $\pm$ indicates the difference from the upper/lower bound of a naive $95\%$ confidence interval assuming normally distributed errors.

| Model | BA |
| --- | --- |
| BERT-Concat | **86.7** |
| BERT-Merge | 79.9 |
| BERT-Featurediff | 67.8 |
| DeBERTa-Concat | 54.7 |
| DeBERTa-Merge | 53.2 |
| DeBERTa-Featurediff | 50.8 |
| RoBERTa-Concat | 52.1 |
| RoBERTa-Merge | 50.3 |
| RoBERTa-Featurediff | 51.1 |
| BERT-Large-Concat | 84.4 |
| BERT-Large-Merge | 84.1 |
| BERT-Large-Featurediff | 59.2 |
| BERT-Bilinear | 50.7 |

## D.1 Synthetic Data

For active learning, we freeze the underlying BERT model during the active learning selection and only apply MC-Dropout on the level of the classifier head, similar to [14], but unlike them we do not use BALD [42] and instead approximate $p(y|s, s')$ averaging the models' predicted probabilities $p_{\hat{\varphi}}(y|s, s', w)$ for 50 sampled dropout masks $w$. We call this approach LC-UNC and experimented with various alternative selection criteria. Unlike LC-UNC, LC directly approximates $1 - \max_y p(y|s, s')$ using a single forward pass through the $\hat{\varphi}$ with deactivated dropout. BALD is the approach from [14], while VARRA and Majority approximate $1 - \max_y p(y|s, s')$ using MC-Dropout differently than LC-UNC: In Majority, $p(y|s, s')$ is approximated as the fraction of dropout samples $w$ for which $\hat{\varphi} = 1$, while VARRA averages $1 - \max_y p_{\hat{\varphi}}(y|s, s', w)$ over dropout samples $w$ instead of averaging $p_{\hat{\varphi}}(y|s, s', w)$ before applying the maximum operator. In addition, the table contains the "automatic relabeling" condition in which $D_i$ is selected from the whole of $C^e$ rather than just the previously unlabeled examples $D_i \subset C^e \setminus \bigcup_{j<i} D_j$. During training, pairs $(s, s')$ that have been queried multiple times are labelled according to the majority vote of all queries, and as $0.5$ in case of a tie.

We validate the efficacy of our active learning approach for learning the similarity function $\varphi(s, s')$ with a limited amount of noisy queries. For this, we define two synthetic similarity functions $\varphi_i : i \in \{1, 2\}$. The first, $\varphi_1$ is equal to zero, whenever a pair $(s, s')$ was generated via word replacement and equal to one otherwise, as in the first experiment from the previous section. The second, $\varphi_2$ is equal to zero, whenever the group $j$ of $s$ that was removed and the added group $j'$ in $s'$ are within the same category of gender and sexuality, race, or religion, and equal to one otherwise. For example, a pair $(s, s')$ for which markers of "White people" in $s$ were modified to markers of "Black people" in $s'$ would have $\varphi_2(s, s') = 0$, while $\varphi_2(s, s')$ would be one if the group was modified to "muslim" in $s'$ instead. We simulate the label noise introduced by annotators' disagreement by independently flipping each label with probability $p = 0.3$ during training the similarity classifier $\hat{\varphi}$. For training with 3 instead of one query per data point, we reduce the overall amount of training data from 10000 samples in $C^e$ to 3333 samples and reduce the probability of flipping labels to $p = 0.216$, simulating a majority vote. In turn, the active learning approach selects 333 instead of 1000 data points for labeling in each of its ten steps in that scenario. Table D.2 shows that active learning

Table D.2: Balanced accuracy for BERT classifier trained using a constant amount of 50k gradient steps and a constant amount of 10k queries. All results are averaged over 10 runs and $\pm$ indicates the difference from the upper/lower bound of a naive 95% confidence interval assuming normally distributed errors.

| Method/Dataset | $\varphi_2$ (Same category) | $\varphi_1$ (Word replacement) |
|---|---|---|
| Random sampling, 1 query | $75.1 \pm 3.6$ | $74.8 \pm 1.8$ |
| Random sampling, 3 queries | $71.6 \pm 3.9$ | $72.5 \pm 1.5$ |
| Random sampling, 5 queries | $70.7 \pm 2.7$ | $73.4 \pm 1.8$ |
| BALD 1 query | $75.9 \pm 4.0$ | $77.9 \pm 2.1$ |
| BALD 3 queries | $73.8 \pm 6.5$ | $78.1 \pm 1.7$ |
| BALD automatic relabeling | $76.1 \pm 4.5$ | $77.6 \pm 2.6$ |
| LC 1 query | $79.1 \pm 4.4$ | $78.5 \pm 1.8$ |
| LC 3 queries | $74.6 \pm 2.4$ | $79.5 \pm 1.8$ |
| LC automatic relabeling | $73.4 \pm 5.9$ | $78.2 \pm 1.3$ |
| LC-UNC 1 query | $79.0 \pm 4.9$ | $79.7 \pm 1.5$ |
| LC-UNC 3 queries | $75.8 \pm 5.4$ | $78.7 \pm 2.6$ |
| LC-UNC automatic relabeling | $76.6 \pm 3.9$ | $76.7 \pm 1.5$ |
| VARRA 1 query | $77.3 \pm 7.4$ | $78.9 \pm 2.1$ |
| VARRA 3 queries | $73.1 \pm 5.7$ | $79.8 \pm 1.6$ |
| VARRA automatic relabeling | $77.7 \pm 2.9$ | $78.0 \pm 1.3$ |
| Majority 1 query | $74.9 \pm 3.5$ | $76.8 \pm 2.4$ |
| Majority 3 queries | $78.7 \pm 5.2$ | $79.6 \pm 0.9$ |
| Majority automatic relabeling | $74.4 \pm 6.2$ | $77.9 \pm 1.8$ |

noticeably outperforms randomly sampling data points for our task, that there is no clear direct benefit from employing multiple queries per pair $(s, s') \in C^e$ over obtaining labels for previously unseen pairs, and that the LC-UNC setup is usually performing as well as or better than alternative selection criteria in the one-query per data point setting.

## D.2  Human Evaluation

Tables D.3 and D.4 show additional results on the active learning from human feedback. As above, we tested our approach using different filtering thresholds $t$ on the two test sets $T$ (Table D.3) and $S$ (Table D.4). In the Retrain condition, the classifier $\hat{\varphi}$ was trained for a single epoch on all labeled datapoints $\bigcup_{i<n} D_i$ in order to combat potential issues with catastrophic forgetting. In the Retrain + Reweigh condition, the same was done, but the Cross Entropy loss was reweighed to balance the empirical label frequencies in $\bigcup_{i<n} D_i$. In the From Scratch setting, we train a new classifier on $\bigcup_{i<n} D_i$ for 5 epochs from scratch without first training it separately on any $D_i$. Again, datapoints are reweighed according to their empirical frequency in $\bigcup_{i<n} D_i$ in the From Scratch + Reweigh setting.

# E  Word Lists And Example Generations

Tables E.2–E.4 show 5 randomly example pairs $(s, s')$ produced by our style transfer approach and GPT-3 in zero-shot and edit mode. Warning: Some of the example texts contain offensive language.

Table D.3: Results for active learning to predict human fairness judgments, on test data $T$. Active learning classifiers are retrained 10 times on the last batch $D_6$. Results are averaged and $\pm$ indicates the difference from the upper/lower bound of a naive $95\%$ confidence interval assuming normally distributed errors.

| Method | ACC | TNR | TPR |
|---|---|---|---|
| Baseline: Constant 0 | 78.8 | 100.0 | 0.0 |
| AL t=0.5 | $79.8 \pm 0.3$ | $97.2 \pm 0.3$ | $15.1 \pm 1.2$ |
| AL + Relabel t=0.5 | $81.1 \pm 0.3$ | $95.5 \pm 0.7$ | $28.6 \pm 2.2$ |
| AL + Relabel + Retrain t=0.5 | $79.6 \pm 0.4$ | $95.3 \pm 1.4$ | $21.5 \pm 3.9$ |
| AL + Relabel + Retrain + Reweigh t=0.5 | $79.6 \pm 0.8$ | $93.9 \pm 1.6$ | $26.6 \pm 3.4$ |
| From Scratch t=0.5 | $77.5 \pm 1.3$ | $90.8 \pm 3.3$ | $28.1 \pm 7.1$ |
| From Scratch + Reweigh t=0.5 | $77.7 \pm 1.4$ | $91.0 \pm 2.7$ | $28.3 \pm 5.0$ |
| AL t=0.1 | $80.0 \pm 0.5$ | $95.2 \pm 0.7$ | $23.7 \pm 3.5$ |
| AL + Relabel t=0.1 | $80.7 \pm 0.6$ | $93.0 \pm 0.9$ | $35.0 \pm 1.3$ |
| AL + Relabel + Retrain t=0.1 | $62.1 \pm 5.6$ | $61.5 \pm 8.9$ | $64.0 \pm 7.0$ |
| AL + Relabeling + Retrain + Reweigh t=0.1 | $52.8 \pm 6.2$ | $46.8 \pm 7.7$ | $75.0 \pm 4.6$ |
| From Scratch t=0.1 | $53.4 \pm 7.9$ | $48.6 \pm 14.3$ | $71.1 \pm 9.2$ |
| From Scratch + Reweighed t=0.1 | $54.8 \pm 6.7$ | $51.2 \pm 10.5$ | $67.9 \pm 9.1$ |
| AL t=0.01 | $78.7 \pm 1.1$ | $87.5 \pm 2.1$ | $45, 7 \pm 1.8$ |
| AL + Relabel t=0.01 | $78.3 \pm 0.7$ | $86.8 \pm 1.5$ | $46.6 \pm 2.5$ |
| AL + Relabel + Retrain t=0.01 | $21.2 \pm 0.1$ | $0.0 \pm 0.0$ | $100 \pm 0.0$ |
| AL + Relabel + Retrain + Reweigh t=0.01 | $21.1 \pm 0.0$ | $0.0 \pm 0.0$ | $100 \pm 0.0$ |
| From Scratch t=0.01 | $21.7 \pm 0.5$ | $0.0 \pm 0.0$ | $99.5 \pm 0.6$ |
| From Scratch + Reweigh t=0.01 | $21.8 \pm 1.5$ | $1.5 \pm 3.6$ | $98.3 \pm 1.7$ |

Table D.4: Results for active learning to predict human fairness judgments, using the separate test data $S$. Active learning classifiers are retrained 10 times on the last batch $D_6$. Results are averaged and $\pm$ indicates the difference from the upper/lower bound of a naive $95\%$ confidence interval assuming normally distributed errors.

| Method | ACC | TNR | TPR |
|---|---|---|---|
| Baseline: Constant 0 | 96.1 | 100.0 | 0.0 |
| AL t=0.5 | $93.8 \pm 0.5$ | $97.0 \pm 0.6$ | $14.6 \pm 2.2$ |
| AL + Relabel t=0.5 | $92.1 \pm 0.6$ | $95.1 \pm 0.7$ | $18.9 \pm 2.7$ |
| AL + Relabel + Retrain t=0.5 | $90.7 \pm 1.7$ | $93.8 \pm 1.9$ | $12.8 \pm 4.0$ |
| AL + Relabel + Retrain + Reweigh t=0.5 | $89.0 \pm 1.3$ | $92.0 \pm 1.4$ | $16.4 \pm 3.4$ |
| From Scratch t=0.5 | $89.2 \pm 2.6$ | $91.8 \pm 2.5$ | $25.7 \pm 5.5$ |
| From Scratch + Reweigh t=0.5 | $89.2 \pm 2.5$ | $91.8 \pm 2.7$ | $25.7 \pm 4.4$ |
| AL t=0.1 | $90.4 \pm 1.3$ | $93.3 \pm 1.3$ | $21.0 \pm 2.3$ |
| AL + Relabel t=0.1 | $89.6 \pm 0.8$ | $92.2 \pm 0.8$ | $24.6 \pm 1.4$ |
| AL + Relabel + Retrain t=0.1 | $60.0 \pm 8.1$ | $59.5 \pm 8.8$ | $72.8 \pm 11.9$ |
| AL + Relabel + Retrain + Reweigh t=0.1 | $46.7 \pm 7.4$ | $45.2 \pm 8.0$ | $83.9 \pm 7.6$ |
| From Scratch t=0.1 | $50.6 \pm 10.4$ | $49.8 \pm 11.2$ | $69.6 \pm 9.3$ |
| From Scratch + Reweigh t=0.1 | $55.0 \pm 9.4$ | $54.5 \pm 10.0$ | $66.7 \pm 6.6$ |
| AL t=0.01 | $80.6 \pm 2.3$ | $82.3 \pm 2.7$ | $38.2 \pm 6.8$ |
| AL + Relabel t=0.01 | $80.2 \pm 1.3$ | $85.5 \pm 1.4$ | $30.0 \pm 2.7$ |
| AL + Relabel + Retrain t=0.01 | $3.9 \pm 0.0$ | $0.0 \pm 0.0$ | $100.0 \pm 0.0$ |
| AL + Relabel + Retrain + Reweigh t=0.01 | $3.9 \pm 0.0$ | $0.0 \pm 0.0$ | $100.0 \pm 0.0$ |
| From Scratch t=0.01 | $4.6 \pm 0.9$ | $0.0 \pm 0.1$ | $99.6 \pm 0.4$ |
| From Scratch + Reweigh t=0.01 | $5.4 \pm 3.9$ | $1.6 \pm 3.2$ | $50.8 \pm 1.6$ |

| Demographic Group | Descriptors | Nouns |
|---|---|---|
| Male | male,manly,masculine | man, men, grandfather, bro, guy, boy, father, dad, son, husbands, husband, grandpa, brother |
| Female | female, pregnant, feminine, femme, womanly | woman, women, grandmother, lady, ladies, girl, mother, mom, daughter, wives, wife, grandma, sister |
| Transgender | transsexual, FTM, F2M, MTF, trans, M2F, transgender, trans female, trans feminine, trans femme,trans fem, trans male,trans masculine,trans masc | |
| Heterosexual | straight, heterosexual, hetero | |
| Homosexual | gay, homosexual, lesbian | |
| Christian | Amish, Catholic, Christian, Evangelical, Lutheran, Mormon, Protestant, Unitarian | |
| Jewish | Conservative Jewish, Orthodox Jewish, Reform Jewish, Hasidic, Jewish | |
| Muslim | Muslim, Shia, Sunni | |
| Hindu | Hindu | |
| Buddhist | Buddhist | |
| Atheist | atheist, irreligious | |
| Black | African-American, African, Black, BIPOC | |
| White | European-American, European, Caucasian, White | |
| Asian | Chinese-American, American-Born Chinese, Chinese, Indian-American, Indian, Korean-American, Korean, Vietnamese-American, Vietnamese, AAPI | |
| Latino | Cuban-American,Cuban, Dominican-American, Dominican, Salvadoran-American, Salvadoran, Guatemalan-American, Guatemalan, Mexican-American, Mexican, Filipina-American, Filipina, Filipino-American, Filipino, Hispanic, Latinx, Latine, Latino, Latina, Latin American | |

Table E.1: Lists of words used for word replacementm, based on Smith et al. [20]

| Original Comment | Modified Comment |
| --- | --- |
| 'Hahaha, really? Thousands of years of war and it was mostly white men, huh? Read some history about non-European cultures to pierce your idyllic bubble.' | 'Hahaha, really? Thousands of years of war and it was mostly white European Christians, huh? Read some history about non-European cultures to pierce your idyllic bubble' |
| 'I would really like to see your source for the Japanese fear of the Russians as being any kind of factor whatsoever. For most of the Japanese people, the Russians were not even on the radar screen!!' | 'I would really like to see your source for the Buddhist fear of the west as being any kind of factor whatsoever. For most of the western people, the west and religion were not even on the radar screen!!' |
| 'Yes: at least implicitly it endorses the old canard that the United States is a "Christian nation."' | 'Yes: at least implicitly it endorses the old canard that the United States is a "white racist nation."' |
| 'Please be careful what you label fake news. News of shootings (Greenville?), migrants in Sweden raising crime rates, Hispanics being rapists and murderers, illegal voting, etc those are examples of fake news....feet on couch? not so much....' | 'Please be careful what you label fake news. Hundreds of shootings (Green Mountain?), migrants in Sweden raising crime rates, homosexuals being raped and murderers, illegal voting, etc those are examples of fake news....feet on couch? not so much.' |
| 'And often they are not so silent. Gay friends have told me about various such clergy who are well known – only no one speaks out about it.' | 'And often they are not so silent. Gay friends have told me about various such men who are well known – only no one speaks out about it.' |

Table E.2: Examples of pairs produced by our style transfer approach

| Original Comment | Modified Comment |
| --- | --- |
| 'OK, so now Im confused. Is it or is it not OK to mock a reporters physical handicap? Call a Latina beauty queen "Miss Housekeeping"? Post a mock video showing yourself clocking a woman you dont like with a hard-hit golf ball?' | ' Is it or is it not OK to mock a reporters physical handicap? Call a Muslim beauty queen "Miss Housekeeping"? Post a mock video showing yourself clocking a woman you dont like with a hard-hit golf ball?' |
| "You caught me. I confess. I've been lying this whole time. I'm not actually an atheist, of course I believe in a god. I see him in the mirror every day !" | " You caught me. I confess. I've been lying this whole time. I'm not actually a homosexual, of course I believe in a god. I see him in the mirror every day !" |
| "Denying the existence of racism in America,(or most anywhere), is just ignoring reality. The formal dicrimination has largely been curbed(with exception of using race to design Congressional districts), but those wonderful 'family values' of the white South still carry the mythology of inherent white superiority" | ". Denying the existence of racism in America,(or most anywhere), is just ignoring reality. The formal dicrimination has largely been curbed(with exception of using race to design Congressional districts), but those wonderful 'family values' of the Hindu South still carry the mythology of inherent Hindu" |
| 'Cite to "the Republicans, Trump being a Republican, who are against gays" please. New one to me. (Other than Trump being registered Republican of course, lol!)' | ' Christian people are against gays.' |
| "You'd better watch out if you don't claim Christianity unto yourself like they do. It is true that people harbor resentment over people whom they view as different. It's the only reason I can think of why they hate gay people." | " You'd better watch out if you don't claim Christianity unto yourself like they do. It is true that people harbor resentment over people whom they view as different. It's the only reason I can think of why they hate Asians." |

Table E.3: Examples of pairs produced by GPT-3 (zero-shot)

| Original Comment | Modified Comment |
| --- | --- |
| 'So do they cover their faces? This is about being able to identify women for practical purposes. Makes sense to moi.' | 'So do they cover their faces? This is about being able to identify White people for practical purposes. Makes sense to moi. ' |
| 'What about LGBTQI+M(xn) W(omxn)Y(oxth)C(hxldrxn)O(thxrs)' | 'Christianity is the best religion. Christian people are the best people. Jesus Christ was a man who lived a perfect life and died for our sins. We should all be following Jesus because he is the author and finisher of our faith. ' |
| 'Bust Judaism, Christianity and Islam are all Abrahamic religions tracing their origins to the patriarch Abraham. They are all related.' | 'Bust Hispanic people are all the same. They are all related. ' |
| "AO96 doesn't allow men into womens rest rooms or locker rooms. You do realize that trans people have been using the bathroom with you forever right? Have you ever been injured by any of these people in all the years you have been using public facilities?" | "AO96 doesn't allow white people into black rest rooms or locker rooms. You do realize that trans people have been using the bathroom with you forever right? Have you ever been injured by any of these people in all the years you have been using public facilities? " |
| 'so.. I . have seen dozens of women in niqab in Toronto this week alone Almost every day I see groups of them. The problem with the niqab policy is that is was for such a silly narrow issue. The niaqb should have been banned from public buildings and shopping malls.' | 'so.. I . have seen dozens of black people in the niqab in Toronto this week alone Almost every day I see groups of them. The problem with the niqab policy is that is was for such a silly narrow issue. The niaqb should have been banned from public buildings and shopping malls. ' |

Table E.4: Examples of pairs produced by GPT-3 (edit mode)

