# OpenReview forum: "Generating Intuitive Fairness Specifications for Natural Language Processing"
_NeurIPS.cc/2022/Workshop/TSRML — TSRML2022_

### Official Review · Reviewer_whYh · 2022-10-16
**A good paper, making sufficient contributions.**

**Overall Recommendation:** Accept. Details are referred to above…
**Overall Rating:** 7

**Summary:**

Summary.

This paper is dedicated to designing generative intuitive fairness specifications for natural language processing. They point out the gap between human intuition about these perturbations and the formal similarity specifications. The authors leverage unsupervised style transfer and GPT-3's zero-shot capabilities to generate expressive candidate pairs of semantically similar sentences.

**Strengths:**

Pros.

1. I acknowledge the contribution of this paper's specification to the fairness community.

2. Extensive empirical are conducted.

3. The paper is well-written and easy to follow.

**Weaknesses:**

Cons.

1. Why table 2 and the last column in table 3 do not have the error bar?

2. As shown in Table 3, the relabel technique only has marginal benefits, since the error bar is highly overlapped.

**Review Confidence:**

2: The reviewer is willing to defend the evaluation, but it is quite likely that the reviewer did not understand central parts of the paper

---

### Official Review · Reviewer_wmCj · 2022-10-17
**Import Direction & Well Executed Paper**

**Overall Recommendation:** Good paper!
**Overall Rating:** 7

**Summary:**

This paper introduces a technique to generate similar sentence pairs that vary along a sensitive attribute for the purposes of evaluating fairness. Their method relies on LLM's zero shot style transfer capabilities, replacing demographic attributes using hardcoded rules and expaning on them with GPT3. They also validate their method using human studies where they ensure the sentences mean similar things but differ along a sensitive attribute.

**Strengths:**

- Important problem, well written, well executed design and study


**Weaknesses:**

I don't have any strong weaknesses with the work to raise.

It would be interesting to apply this technique for further problems (NLI, QA, etc.)  in the future.

The applications for studying fairness are interesting, but I'm not entirely sure the method needs to be just fairness specific. Could this technique reduce any "bias" in the dataset (e.g., any artifact that might exist?)

**Review Confidence:**

4: The reviewer is confident but not absolutely certain that the evaluation is correct

---

### Decision · Program_Chairs · 2022-10-23

Accept